# Mesenchymal Stem Cell Derived Biocompatible Membrane Vesicles Demonstrate Immunomodulatory Activity Inhibiting Activation and proliferation of Human Mononuclear Cells

**DOI:** 10.3390/pharmaceutics12060577

**Published:** 2020-06-23

**Authors:** Marina O. Gomzikova, Sevindzh K. Kletukhina, Sirina V. Kurbangaleeva, Olga A. Neustroeva, Olga S. Vasileva, Ekaterina E. Garanina, Svetlana F. Khaiboullina, Albert A. Rizvanov

**Affiliations:** 1Institute of Fundamental Medicine and Biology, Kazan Federal University, Kazan 420008, Russia; SKKletuhina@kpfu.ru (S.K.K.); SiVKurbangaleeva@kpfu.ru (S.V.K.); OlANeustroeva@kpfu.ru (O.A.N.); olyavas@bk.ru (O.S.V.); EEGaranina@kpfu.ru (E.E.G.); sv.khaiboullina@gmail.com (S.F.K.); Albert.Rizvanov@kpfu.ru (A.A.R.); 2M.M. Shemyakin–Yu.A. Ovchinnikov Institute of Bioorganic Chemistry of the Russian Academy of Sciences, Moscow 117997, Russia; 3Department of Microbiology and Immunology, University of Nevada, Reno School of Medicine, Reno, NV 89557, USA

**Keywords:** extracellular vesicles, microvesicles, cytochalasin B-induced membrane vesicles, mesenchymal stem cells, immunosuppression, immunomodulation, PBMCs, mononuclear cells

## Abstract

Immune-mediated diseases are characterized by abnormal activity of the immune system. The cytochalasin B-induced membrane vesicles (CIMVs) are innovative therapeutic instruments. However, the immunomodulating activity of human mesenchymal stem cell (MSC)-derived CIMVs (CIMVs-MSCs) remains unknown. Therefore, we sought to investigate the immunological properties of CIMVs-MSCs and evaluate their effect on human peripheral blood mononuclear cells (PBMCs). We found that CIMVs-MSCs are primarily uptaken by monocytes and B-cells. Additionally, we demonstrated that CIMVs-MSCs inhibit phytohemagglutinin (PHA)-induced proliferation of PBMCs, with more pronounced effect on T-lymphocytes expansion as compared to that of B-cells. In addition, activation of T-helpers (CD4+CD25+), B-cells (CD19+CD25+), and T-cytotoxic lymphocytes (CD8+CD25+) was also significantly suppressed by CIMVs-MSCs. Additionally, CIMVs-MSCs decreased secretion of epidermal growth factor (EGF) and pro-inflammatory Fractalkine in a population of PBMCs, while the releases of FGF-2, G-CSF, anti-inflammatory GM-CSF, MCP-3, anti-inflammatory MDC, anti-inflammatory IL-12p70, pro-inflammatory IL-1b, and MCP-1 were increased. We analyzed the effect of CIMVs-MSCs on an isolated population of CD4+ and CD8+ T-lymphocytes and demonstrated their different immune response and cytokine secretion. Finally, we observed that no xenogeneic nor allogeneic transplantation of CIMVs induced an immune response in mice. Our data suggest that CIMVs-MSCs have immunosuppressive properties, are potential agents for immunomodulating treatment, and are worthy of further investigation.

## 1. Introduction

Immune-mediated diseases such as autoimmune diseases, allergies and graft-versus-host reaction are serious healthcare challenges worldwide. Current treatment strategies of these diseases have multiple weaknesses and deficiencies, urging the development of innovative immune-modulation therapeutics. Mesenchymal stem cell (MSC)-based therapies are an effective approach for a range of immune-mediated diseases, including chronic autoimmune urticaria, multiple sclerosis, Crohn’s disease, rheumatoid arthritis, systemic lupus erythematosus, and many others [1]. However, the safety, specific procedure of therapeutic preparation, and special requirements for storage and transportation restrict clinical use of MSCs [2].

It is believed that the therapeutic efficacy is mediated by the paracrine action of extracellular vesicles (EVs) released by MSCs [3]. Therefore, a cell-free treatment approach using EVs was suggested as an alternative and demonstrated better biodistribution as well as no risk of carcinogenesis or blood vessels occlusion [2,4]. Additionally, EVs can be stored until use [2]. MSC-derived EVs retain the anti-inflammatory property of the parental cells and have shown immunosuppressive effects on dendritic cells, T cells, B-cells, and macrophages [5,6]. Currently, human MSC-derived EVs have been used in two clinical cases where significant improvements of symptoms were demonstrated [7,8].

The major obstacle in clinical use of EVs is limited yield [9]. To increase EVs production, the cytochalasin B-induced membrane vesicles (CIMVs) approach was suggested, which significantly improved the yield of vesicles, making large-scale manufacturing feasible [1,10]. Using this approach, CIMVs were derived from HEK293 [10,11,12], 3T3 fibroblast [11], human umbilical vein endothelial cells (HUVECs) [10], MDCKII-MDR1 [13], SH-SY5Y [14], PC3 cells [15], and MSCs [16]. More so, CIMVs were used as vectors for nanoparticles and drug delivery [11,17]. Previously, we demonstrated that CIMVs had biological activity of the parental cells inducing angiogenesis in vivo [14]. We demonstrated that CIMVs derived from MSCs (CIMVs-MSCs) retained the parental content including multiple growth factors and cytokines [16]. The main differences of CIMVs from EVs are as follows: (1) more homogeneous content—the isolation method of CIMVs excludes the sorting of molecules; and (2) CIMVs are produced from washed cells, whereas EVs are isolated from body fluids or conditioned medium. CIMVs might imitate natural EVs in size and cytoplasmic membrane to serve as a vector for drug and bioactive molecule delivery.

Recently, we showed that CIMVs-MSCs had immunosuppressive properties by reducing the antibody production and modestly decreasing the leukocyte number and macrophage phagocytic activity in mice [18]. To investigate the mechanism of CIMVs-MSCs immunosuppressive effect, we analyzed the effect of human CIMVs-MSCs on peripheral blood mononuclear cells (PBMCs). The objectives of this study were to determine the effect of CIMVs-MSCs on leukocyte proliferation, expression of activation markers, and cytokine release. Because xenotransplantation of human cells and extracellular vesicles has been applied in numerous exploratory and preclinical studies, we analyzed the immunogenicity of human MSC-derived CIMVs in mice. Additionally, we evaluated the immunogenicity of allogeneic transplantation of murine MSC-derived CIMVs.

## 2. Materials and Methods

### 2.1. PBMC Isolation

Whole-blood samples from healthy donors were collected in sodium citrate tubes (Vacuette, Monroe, NC, USA). Written informed consent was signed and collected from the donors. Peripheral blood mononuclear cells (PBMCs) were isolated from anticoagulated blood via Ficoll (PanEco, Moscow, Russia) gradient centrifugation. PBMCs were maintained in RPMI (PanEco, Moscow, Russia) supplemented with 10% fetal bovine serum (Gibco, UK) and 2 mM L-glutamine (PanEco, Moscow, Russia). PBMCs were maintained at 37 °C, 5% CO2.

Ethics statement—The Institutional Review Board of the Kazan Federal University approved this study, and informed consent was obtained from each study subject according to the guidelines approved under this protocol (article 20, Federal Law “Protection of Health Rights of Citizens of Russian Federation” N323- FZ, 11.21.2011).

### 2.2. FACS-Based Sorting of PBMCs

T-cytotoxic (CD45+CD3+CD8+), T-helper (CD45+CD3+CD4+), and B (CD45+CD3-CD20+) lymphocytes were separated using FACS. PBMCs (1 × 10^6^ cells/mL) were stained with FITC anti-human CD3 Antibody (300312, BioLegend, San Diego, CA, USA), APC anti-human CD4 Antibody (357404, BioLegend, San Diego, CA, USA), PE anti-human CD8 Antibody (2323530, Sony, San Jose, CA, USA), APC/Cy7 anti-human CD14 Antibody (2109100, Sony, San Jose, CA, USA), and Brilliant Violet 421™ Anti-human CD20 Antibody (2111650, Sony, San Jose, CA, USA) following the manufacturer’s recommendations. In 96-well plates, 1.5 × 10^5^ cells per well were seeded. CIMVs-MSCs (10 μg per well) were added 24 h after the sorting.

### 2.3. CFDA SE Staining of PBMC

Cell permeant dye CFDA SE (65-0850-84, eBioscience, San Diego, CA, USA) was used to analyze the leukocyte proliferation according to the manufacturer’s recommendation. Briefly, PBMCs (1 × 10^6^ cells/mL) were incubated with CFDA SE (10 µM; 65-0850-84, eBioscience, San Diego, CA, USA) for 15 min and washed with RPMI supplemented with 10% FBS and 2 mM L-glutamine.

### 2.4. CIMVs Production

CIMVs-MSCs were prepared as described previously [16]. MSCs were isolated from human adipose tissue. Signed informed consent was obtained from all donors. All experiments were carried out in accordance with an experimental protocol approved by the Biomedicine Ethic Expert Committee of Kazan Federal University and Republican clinical hospital (No. 218, 11.15.2012) based on article 20 of the Federal Legislation on “Health Protection of Citizens of the Russian Federation” № 323-FL, 21.11.2011. Adipose tissue was incubated in 0.2% collagenase II (Dia-M, Moscow, Russia) solution for one hour in a shaker-incubator at 37 °C, 120 rpm. Cell suspension was pelleted (400 g for 5 min), washed once in PBS (PanEco, Moscow, Russia), and re-suspended in DMEM (PanEco, Moscow, Russia) supplemented with 10% fetal bovine serum (Gibco, Paisley, UK) and 2 mM L-glutamine (PanEco, Moscow, Russia). MSCs were maintained at 37 °C, 5% CO2 with culture medium replaced every three days. Human MSCs (passage 4) brightly expressing surface markers (CD90+, CD44+, CD29+, CD73+, STRO-1+) were chosen for the CIMVs production.

MSCs cultures were washed twice with PBS and maintained in DMEM supplemented with 10 µg/mL cytochalasin B (C6762, Sigma-Aldrich, St. Louis, MO, USA) for 30 min (37 °C, 5% CO2). MSCs suspension was vortexed vigorously for 30 s and pelleted (100 g for 10 min). The supernatant was collected and subjected to two subsequent centrifugation steps (300 g for 20 min and 2000 g for 25 min). The CIMVs-MSC pellet was washed once (PBS, 2000 g for 25 min) before use.

### 2.5. Staining with Calcein AM

Calcein AM was used for the cell cytoplasm staining. MSCs (1 × 10^6^ cells/mL) were incubated in DPBS containing 10 µM of Calcein AM (eBioscience, San Diego, CA, USA) for 15 min and washed (1×) with complete medium (DMEM with 10% EV-depleted FBS, 2 mM L-glutamine). Hoechst 33342 (Cat. No. sc-200908, Santa Cruz, USA) was used for the nuclei staining. Cells were incubated in DPBS containing Hoechst 33342 for 10 min at room temperature and washed (1×) with DPBS.

### 2.6. Flow Cytometry Analysis with Calibration Particles

Pellets after each step of sequential centrifugation were analyzed by flow cytometry. A mixture of calibration particles 1.34, 3.4, 5.1, and 14.5 μm (Cat. No. PPS-6K, Spherotech, Lake Forest, IL, USA) was used for calibration of BD FACS Aria III (BD Bioscience, San Jose, CA, USA). Then, the sizes of native cells and cell components in the pellet after the first step of centrifugation (100 g for 10 min), in the pellet after the second step of centrifugation (300 g for 20 min), and in the pellet after the last step of centrifugation (2000 g for 25 min) were analyzed. Histograms were built using BD FACSDiva 8 software (BD Bioscience, San Jose, CA, USA).

### 2.7. Scanning Electron Microscopy (SEM)

CIMVs-MSCs were fixed (10% formalin for 15 min), dehydrated using graded alcohol series, and dried at 37 °C. Prior to imaging, samples were coated with gold/palladium in a Quorum T150ES sputter coater (Quorum Technologies Ltd., Lewes, United Kingdom). Slides were analyzed using a Merlin field emission scanning electron microscope (Carl Zeiss, Oberkochen, Germany).

### 2.8. Staining of CIMVS with DiD

Lipophilic DiD dye (Life Technologies, Carlsbad, CA, USA) was used to trace CIMVs-MSCs uptake by PBMCs in vitro. CIMVs-MSCs suspension (300 μg/mL) was incubated with DiD dye (5 µM) for 15 min (37 °C, 5% CO2) and washed twice with complete medium (DMEM with 10% FBS, 2 mM L-glutamine) before use.

### 2.9. Phytohemagglutinin (PHA) Activation

Phytohemagglutinin (PHA) (PanEco, Moscow, Russia) was used to induce activation and proliferation of lymphocytes in vitro. PBMCs were incubated with CIMVs-MSCs for 24 h in complete medium (RPMI with 10% FBS, 2 mM L-glutamine) followed by incubation with 10 μg/mL PHA for three days. Three days later, PBMCs were washed once with PBS and stained with monoclonal FITC anti-human CD3 antibodies (300312, BioLegend, San Diego, CA, USA), APC anti-human CD4 antibodies (357404, BioLegend, San Diego, CA, USA), PE anti-human CD8 antibodies (2323530, Sony, San Jose, CA, USA), APC/Cy7 anti-human CD14 antibodies (2109100, Sony, San Jose, CA, USA), Brilliant Violet 421™ anti-human CD20 antibodies (2111650, Sony, San Jose, CA, USA), Alexa Fluor^®^ 647 anti-human CD56 antibodies (2191565, Sony, San Jose, CA, USA), and APC anti-human CD19 antibodies (IM2470, Beckman Coulter, Brea, CA, USA) following the manufacturer’s recommendations.

### 2.10. Multiplex Analysis

Multiplex analysis based on xMAP Luminex technology was done using Premixed 41 Plex Immunology Multiplex Assay (sCD40L, EGF, Eotaxin/CCL11, FGF-2, Flt-3 ligand, Fractalkine, G-CSF, GM-CSF, GRO, IFN-α2, IFN-γ, IL-1α, IL-1β, IL-1ra, IL-2, IL-3, IL-4, IL-5, IL-6, IL-7, IL-8, IL-9, IL-10, IL-12 (p40), IL-12 (p70), IL-13, IL-15, IL-17A, IP-10, MCP-1, MCP-3, MDC (CCL22), MIP-1α, MIP-1β, PDGF-AA, PDGF-AB/BB, RANTES, TGF-α, TNF-α, TNF-β, VEGF) (Merckmillipore, USA), according to manufacturer’s instructions. Briefly, samples were incubated with fluorescent beads for 1 h at room temperature (RT) and washed and incubated with phycoerythrin–streptavidin for 10 min at RT (Merckmillipore, USA). Data were analyzed using MasterPlex CT control software and MasterPlex QT analysis software (MiraiBio, San Bruno, CA, USA). Culture media of cells (PBMCs, T-cytotoxic, T-helper cells, or B-lymphocytes) treated with CIMVs-MSCs or untreated were used for multiplex analysis.

### 2.11. Animals

Adult mice (*Mus musculus*, BALB/c) (Pushchino, Russia) were used for the experiments. All experiments were carried out in compliance with the procedure protocols approved by Kazan Federal University local ethics committee (protocol #5, date 27.05.2014) according to the rules adopted by Kazan Federal University and Russian Federation Laws. For immune response analysis, mice received intravenous infection of 15 µg of CIMVs-hMSCs or 15 µg of CIMVs-mMSCs. Each experimental group contained five animals. Mice were euthanized in compliance with the procedure protocols approved by Kazan Federal University local ethics committee (protocol #5, date 27.05.2014).

### 2.12. Immunostaining of Murine PBMCs

Blood was collected in a tube with 3.8% *w/v* sodium citrate. Mouse PBMCs were purified from whole blood by Ficoll (PanEco, Moscow, Russia) density centrifugation. Murine PBMCs staining was performed using the following monoclonal antibodies: CD45-PerCP (103130; BioLegend, San Diego, CA, USA), CD3-PE (100308; BioLegend, San Diego, CA, USA), CD8a-PE/Cy7 (100722; BioLegend, San Diego, CA, USA), CD4-APC (100412; BioLegend, San Diego, CA, USA), CD19-FITC (152404; BioLegend, San Diego, CA, USA), and CD25-Pacific Blue (102022; BioLegend, San Diego, CA, USA).

### 2.13. Statistical Analysis

Statistical analysis was done using Wilcoxon signed-rank test (R-Studio) with significance level *p* ≤ 0.05. Illustrations were built with the “ggplot2” package.

## 3. Results

### 3.1. CIMVs Isolation Procedure

CIMVs were characterized using fluorescence microscopy and flow cytometry. The procedure consisted of three main steps: treatment of cells with cytochalasin B, induction of membrane vesicles pinching off by vortexing, and isolation of CIMVs by sequential centrifugation or by filtration using filters up to 2000 nm pore diameter (Figure 1). The described procedure potentially might be applied to produce CIMVs from any cell containing an actin cytoskeleton.

In our study, we isolated CIMVs by sequential centrifugation to characterize step-by step the sedimented cell components. We found that after the first centrifugation step, cells (double-positive staining with Calcein AM and Hoechst 33342) were sedimented (Figure 2A–C). The most numerous group (5.1–14.5 µm) in size included reduced cells or cytoplasm-depleted cells and karioplasts—nuclei with a small amount of cytoplasm (Figure 2A–C,J; red arrow). It is known that cytochalasin B induces nuclei extrusion by forming karioplasts, which are nuclei enclosed in cytoplasmic membrane (CPM) [19]. Cell components 1.34-3.4 μm in size were large membrane vesicles which also have been detected in the pellet (Figure 2A–C,J). In the pellet after the second step of centrifugation, there were increased amounts of cytoplasm-depleted cells (Figure 2D–F,K; yellow arrow), karioplasts (Figure 2D–F,K; red arrow), and large membrane vesicles (Figure 2D–F,K).

The last centrifugation step led to sedimentation of membrane vesicles which demonstrated staining with a cytoplasmic dye and no nuclei (Figure 2G–I,L). We previously determined that CIMVs-MSCs have sizes ranging from 100 to 2600 nm with the majority (89.36%) having sizes between 100 and 1200 nm [16].

### 3.2. CIMVs Uptake by Leukocytes

CIMVs-MSCs were stained with membrane dye DiD (V22887, ThermoFisher, USA) for 24 h at RT before incubation with PBMCs. Flow cytometry analysis revealed that monocytes (99.5 ± 0.26%; CD14+ cells), NK-cells (29.6 ± 3.2%; CD3-CD56+), B-cells (69.43 ± 9.52%; CD3-CD20+), T-cytotoxic lymphocytes (35.6 ± 3.83%; CD3+CD8+), and T-helper lymphocytes (14.5 ± 4.42%; CD3+CD4+) were positive for DiD dye (Figure 3). We analyzed the CIMVs uptake by confocal microscopy and found co-localization of DiD-stained CIMVs and immune cells stained with antibody (Appendix A; arrows point to co-localization). These data demonstrate that monocytes and B-cells are the most effective at uptaking CIMVs-MSCs, while T-helpers have the least ability to uptake membrane vesicles.

### 3.3. CIMVs-MCSs Inhibition of PHA-Induced Proliferation of PBMCs

PBMCs were incubated with CFDA SE (65-0850-84, ThermoFisher, USA), which allows detection of changes in lymphocyte counts. To analyze the effect of CIMVs-MCSs on lymphocyte proliferation, PBMCs were incubated with CIMVs-MSCs for 24 h and then were treated with 10 μg/mL PHA (M021, PanEco, Moscow, Russia). Three days later, PBMCs were analyzed using flow cytometry (Figure 4A). In control PMBCs (untreated with CIMVs-MSCs and PHA), the proportion of proliferating cells was 9 ± 1.2%, while it was 6.5 ± 2% in CIMVs-MSCs treated PBMCs, which did not differ significantly from that in the control (Figure 4B). As expected, incubation with PHA increased PBMCs proliferation (43.1 ± 5.9%), which was significantly higher than that in the un-stimulated control (*p* = 0.00034) and cells incubated with CIMVs-MSCs (*p* = 0.00022) (Figure 4B). Interestingly, CIMVs-MSCs significantly decreased PHA-induced PBMCs proliferation (15.35 ± 0.6%) (*p* = 0.0041) (Figure 4B).

Next, the inhibitory effects of allogeneic CIMVs-MSCs on PHA-activated proliferation of T-cytotoxic, T-helper, and B-cells were analyzed. We used CFDA SE-stained PBMCs to analyze the effect of CIMVs-MSCs. PBMCs were pretreated with CIMVs-MSCs for 24 h before incubation with PHA. Three days later, the percent of proliferating T-helper (CD45+CD3+CD4+), T-cytotoxic (CD45+CD3+CD8+), and B-cells (CD45+CD3-CD19+) was determined (Figure 5). Increased populations of proliferating CD4+, CD19+, and CD8+ cells were found in CIMVs-MSCs-treated PBMCs (Figure 5). CIMVs-MSCs treatment increased the percent of proliferating CD4+ cells (*p* = 0.0016), CD19+ cells (*p* = 0.008), and CD8+ cells (*p* = 0.04).

PHA significantly increased the percentage of proliferating CD4+ (45.6 ± 11.6% (*p* = 0.0013)), CD19+ (42.23 ± 8.42% (*p* = 0.0006)), and CD8+ (39.33 ± 4.56% (*p* = 0.00006)) leukocytes by 91.2, 18.6, and 39.3 times, respectively (Figure 5). Interestingly, an inhibitory effect of CIMVs-MSCs on PHA-induced proliferation of CD4+, CD19+, and CD8+ cells was demonstrated, where the percent of proliferating cells decreased from 45.6 ± 11.6% to 15.9 ± 1.31% in CD4+ cells (*p* = 0.006), from 42.23 ± 8.42% to 20.53 ± 1.9% in CD19+ cells (*p* = 0.006), and from 39.33 ± 4.56% to 13.4 ± 1.31% in CD8+ (*p* = 0.00035) (Figure 5). Pretreatment of PBMCs with CIMVs-MSCs led to suppression of CD4+, CD19+, and CD8+ cell proliferation by 2.9, 2.1, and 2.9 times, respectively (Figure 5).

We sought to determine the impact of CIMVs uptake efficiency on the proliferation of lymphocytes. We found that 48.7 ± 3.5% of native CD8-positive cells captured CIMVs-MSCs, whereas PHA activation led to 99.76 ± 0.11% of CD8-positive cells capturing CIMVs-MSCs. Among CD4-positive cells, 40 ± 11.85% of native cells vs. 99.8 ± 0.058% of PHA-activated cells captured CIMVs-MSCs, and among B-cells 91.5 ± 4.8% of native cells vs. 99.6 ± 0.4% of PHA-activated cells contained CIMVs-MSCs (Appendix A). Therefore, PHA activation led to increased CIMVs uptake efficiency by lymphocytes. Then, we investigated PHA-activated proliferation of lymphocytes, containing CIMVs-MSCs. We confirmed that CIMVs-MSCs inhibited PHA-activated proliferation of lymphocytes (Appendix A).

### 3.4. Immunosuppressive Activity of CIMVs

CD25 are early leukocyte activation markers [20]. Therefore, we sought to determine the changes in the CD25+ leukocyte population after incubation with CIMVs-MSCs. PHA was used as an activation control as it was shown to upregulate CD25 expression [21].

PBMCs were treated with CIMVs-MSCs for 24 h followed by incubation with PHA for three days. The percent of CD25-expressing T-helper (CD4+CD25+), B-cells (CD19+CD25+), and T-cytotoxic lymphocytes (CD8+CD25+) in the PBMCs was determined using flow cytometry (Figure 6). The proportion of CD25-expressing cells did not differ between control PBMCs and after incubation with CIMVs-MSCs (Figure 6). PHA increased the expression of activation marker CD25 on T-lymphocytes and B-cells, similar to what was previously reported [21,22]. Increased percentages of T-helper (CD4+CD25+; 86.6 ± 5.1% (*p* = 0.000005)), B-cells (CD19+CD25+; 90.7 ± 3.41% (*p* = 0.0002)), and T-cytotoxic lymphocytes (CD8+CD25+; 87.36 ± 3.35% (*p* = 0.000001)) were found in PHA-stimulated PBMCs (Figure 6).

Incubation of PBMCs with CIMVs-MSCs before PHA activation inhibited the expressions of CD25+ in T-helper (37.56 ± 9.14% (0.0006)), B-cells (44.93 ± 8.05% (0.0004)), and T-cytotoxic cells (19.96 ± 2.44% (0.000005)) (Figure 6).

### 3.5. Multiplex Analysis

To analyze the effect of CIMVs-MSCs on cytokine secretion by PBMCs we used the xMap Luminex multiplex method (Table 1). Supernatants collected from PBMCs as well as isolated CD4+ and CD8+ cell populations were used for this study. Culture media were collected 48 h after PBMCs incubation with CIMVs-MSCs.

We found that CIMVs-MSCs decreased the release of growth factors EGF and Fractalkine (CX3CL1) in PBMCs, while it increased the secretion of FGF-2, G-CSF, GM-CSF, MCP-3 (CCL7), MDC (CCL22), IL-12p70, IL-1b, and MCP-1 (CCL2). CIMVs-MSCs did not affect the levels of Eotaxin (CCL11), TGF-a, IFN-a2, IL-10, IL-12p40, PDGF-AA, IL-15, sCD40L, IL-1Ra, IL-1a, IL-9, IL-4, IL-5, IL-7, IL-8, IP10 (CXCL10), MIP-1b (CCL4), and TNFa by PBMCs. Levels of FLT-3L, IFN-g, GRO, IL-13, PDGF-AB/BB, IL-17A, IL-2, IL-3, TNFb, and VEGF were below the detection range.

When isolated lymphocyte populations were analyzed, different patterns of cytokine activation emerged. CD4+ and CD8+ were separated by FACS and used to incubate with CIMVs-MSCs. CD4+ cells incubated with CIMVs-MSCs increased secretion of FGF-2, G-CSF, IL-12p70, IL-4, IL- 6, IL-7, IL-8, IP10 (CXCL10), and MCP-1 (CCL2). A somewhat different group of cytokines was produced by CD8+ cells incubated with CIMVs-MSCs. For example, CIMVs-MSCs decreased the level of IL1b in CD8+, while it was not affected in CD4+ lymphocytes. Additionally, increased levels of Eotaxin was found in CIMVs-MSCs-treated CD8+ as compared to CD4+ cells. It should be noted that the levels of FGF-2, G-CSF, IL-12p70, IL-4, IL-6, IL-7, IL-8, IP10 (CXCL10), and MCP-1 (CCL2) were increased in CIMVs-MSCs-treated CD8+ cells, similar to that in CD4+ lymphocytes.

### 3.6. Transplantation of Allogeneic and Xenogeneic CIMVs

In order to evaluate the immune response on transplantation of allogeneic and xenogeneic CIMVs in mice, percentages of the main populations of immune cells were determined using immunostaining and flow cytometry. Human MSCs-derived CIMVs (15 µg/mouse), murine MSCs derived CIMVs (15 µg/mouse), or saline were injected intravenously. Blood was collected 24 after the injection, stained with antibodies, and analyzed using flow cytometry. We found that no human MSCs-derived CIMVs (CIMVs-hMSCs) nor murine MSCs-derived CIMVs (CIMVs-mMSCs) induced proliferation of immune cells in mice, since no increases in percentages of CD45+, CD3+, CD4+, CD8+, CD19+ cells were observed (Figure 7A). CIMVs-hMSCs and CIMVs-mMSCs did not induce activation of T-lymphocytes (CD3+CD25+ cells), T-helper cells (CD4+CD25+), T-cytotoxic lymphocytes (CD8+CD25+), and B-cells (CD19+CD25+) in mice (Figure 7B). CIMVs-hMSCs even inhibited the normal level (relative to control) of activated CD8+CD25+ cells and CD19+CD25+ cells (Figure 7B).

## 4. Discussion

Immunomodulation is an important therapeutic strategy for the management of many immune-mediated diseases [23]. To abstain the MSCs transplantation while retaining their immunosuppressive effect, cell-free based approaches have been developed. CIMVs provide the tool to produce a large quantity of biocompatible membrane vesicles. In our study, we used cytochalasin B to produce CIMVs-MSCs; analyzed their effect on lymphocyte proliferation, activation, and cytokine production in vitro; and evaluated their immunogenicity in vivo.

Here, for the first time, we report that CD14+ (99.5 ± 0.26%) and CD20+ (69.43 ± 9.52%) cells are preferably uptaken by CIMVs-MSCs, while a lower CIMVs uptake was found in CD8+ (35.6 ± 3.83) and CD56+ (30.26 ± 3.44) cells, and even the smallest amount of CIMVs was detected in CD4+ (14.5 ± 4.42) (Figure 3). Previously, Mao Z. et al. have shown that cytochalasin B-induced cell membrane capsules derived from HEK293 and 3T3 cells could be uptaken by macrophages [11]. This preference in CIMVs uptake by CD14+ could be due to the phagocytosis ability of monocytes and macrophages. Recently, phagocytosis was demonstrated in B-cells as well [24]. Khare D. et al. found that MSC-derived exosomes were mainly found in monocytes and B-cell lymphocytes [25]. These data corroborate our results, where we have demonstrated preferable CIMVs-MSCs uptake by CD14+ and CD20+ cells.

Suppression of PHA-induced PBMCs proliferation by MSCs was previously established [26,27]. Therefore, we sought to determine the effect of CIMVs-MSCs on PHA-induced proliferation of PBMCs. Our data show that CIMVs-MSCs suppressed PHA-induced PBMCs proliferation by 2.8 times, from 43.1 ± 5.9% (PHA-activated PBMC) to 15.35 ± 0.6% (CIMVs-MSCs-treated PBMCs) (Figure 4B).

Inhibitory effects of CIMVs-MSCs on PHA-induced proliferation of CD4+, CD19+, and CD8+ cells were demonstrated, where the percent of proliferating cells decreased 2.9, 2.1, and 2.9 times, respectively (Figure 5). Because almost all PHA-activated lymphocytes contained CIMVs (Appendix A), we believe that is reasonable to conclude the impact of CIMVs on PHA-activated proliferation and activation of lymphocytes.

It was shown that MSCs can inhibit activation of immune cells [28] and decrease the expression of CD25 on PHA-treated lymphocytes [29]. We found that CIMVs-MSCs also suppressed the activation and expression of CD25 on CD4+, CD19+, and CD8+ lymphocytes by 2.3, 2, and 4.4 times, respectively. Therefore, we suggest that, just like MSCs, CIMVs-MSCs can inhibit leukocyte proliferation targeting predominantly CD4+ T cells, B-cells, as well as CD8+ T cells. Further investigation using sorted cell populations including T regulatory cells that constitutively express CD25 will be necessary to understand the molecular mechanisms underlaying CIMVs immunosuppression.

MSCs affect PBMCs cytokine production by reducing TNFα, IL-10, and increasing IL-6, G-CSF, and MCP-1 release [30]. In addition, previously we showed that CIMVs-MSCs contain growth factors, cytokines, and chemokines [16]. Therefore, we sought to determine the effect of CIMVs-MSCs on cytokines produced by PBMCs. Multiplex analysis revealed that CIMVs-MSCs induce secretion of FGF-2 (*p* = 0.000002), anti-inflammatory G-CSF (*p* = 0.021), anti-inflammatory GM-CSF (*p* = 0.007), chemokine MCP-3 (*p* = 0.006), anti-inflammatory MDC (*p* = 0.022), anti-inflammatory IL-12p70 (*p* = 0.043), pro-inflammatory IL-1b (*p* = 0.05), and chemokine MCP-1 (*p* = 0.046) in PBMCs, while they inhibit secretion of EGF (*p* = 0.027) and pro-inflammatory Fractalkine (CX3CL1) (*p* = 0.047). Bertolo A. et al. showed that MSCs when co-cultured with PBMCs could increase secretion of IL-6, G-CSF, and MCP-1, while GM-CSF was not affected [30]. We found that, unlike MSCs, CIMVs-MSCs induce secretion of G-CSF, MCP-1, and also GM-CSF. In addition, we have shown that CIMVs-MSCs induce the secretion of IL1b. It is known that few factors, including IL1b, are required for MSCs-induced immunomodulation [31,32].

Next, to characterize the immunomodulating effect of CIMVs-MSCs, we analyzed their effect on cytokines release by isolated CD4+ and CD8+ cells. We found that CIMVs-MSCs induced the secretion of FGF-2, G-CSF, IL-12p70, and MCP-1 in both CD4+ and CD8+ cells, which could explain our observation of an increased production of these cytokines in whole PBMCs. However, lymphocyte populations differed in their cytokine activation. For example, IL-1b and Eotaxin (CCL11) secretions were increased in CIMVs-MSCs-treated T-cytotoxic lymphocytes (CD8+), while they were not affected in CD4+ cells. In contrast, CIMVs-MSCs increased production of IL-5 in T-helper lymphocytes (CD4+). IL-1b is a key pro-inflammatory cytokine that has been implicated in pain, inflammation, and autoimmune conditions [33]. Decreased IL-1b secretion by T-cytotoxic lymphocytes could explain the mechanism of CIMVs-MSCs immunosuppression. IL-5 is also a pro-inflammatory cytokine that increases eosinophil numbers and antibody levels in vivo [34]. We found that CIMVs did not influence the expressions of IL-15, GM-CSF, and TNF-a, which are cytokines produced by Th1 cells, and they induced secretion of G-CSF (*p* = 0.021), one of the cytokines produced by Th17 cells. Therefore, CIMVs-MSCs did not induce Th1 activation and inflammation. Our data corroborate results published by Rozenberg A. et al. who showed that hMSCs inhibit Th1 responses yet induce Th17 responses [35].

We observed that CIMVs-MSCs did not activate the B-cell population (percent of CD19+CD25+ cells was not affected) in PBMCs, probably due to complex interactions between immune cells in the PBMC population; however, its percent was slightly increased in the PBMCs population. Our data corroborate results published by Rasmusson I. et al. [36]. The authors reported a stimulatory effect of MSCs on human B-cells, showing that MSC have the ability to stimulate or suppress antibody secretion depending on the level of stimulus used to trigger B-cells [36].

Allogeneic MSCs are a promising therapy due to their low immunogenicity [37]. We evaluated the immunogenicity of allogeneic CIMVs-mMSCs and xenogeneic CIMVs-hMSCs. We found that no xenogeneic nor allogeneic transplantation of CIMVs induced proliferation and/or activation of immune cells in mice 24 h after the injection. However, further research is needed with a longer observation time to draw final conclusions about the non-immunogenicity of CIMVs.

## 5. Conclusions

We showed that monocytes and B-cells preferably uptake CIMVs-MSCs. In PHA-stimulated PBMCs, CIMVs-MSCs inhibited proliferation of T-cytotoxic lymphocytes and inhibited B-cells and T-helper lymphocytes. Upon CIMVs-MSCs treatment, PBMCs secreted more of FGF-2, G-CSF, GM-CSF, MCP-3 (CCL7), MDC (CCL22), IL-12p70, IL-1b, and MCP-1 (CCL2). Obtained data can be used as a base for developing immunomodulation therapy using CIMVs-MSCs.

## Figures and Tables

**Figure 1 pharmaceutics-12-00577-f001:**
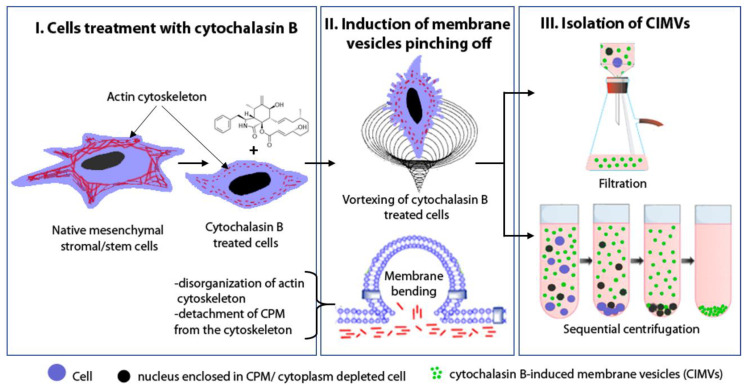
The scheme of cytochalasin B-induced membrane vesicles (CIMVs) production.

**Figure 2 pharmaceutics-12-00577-f002:**
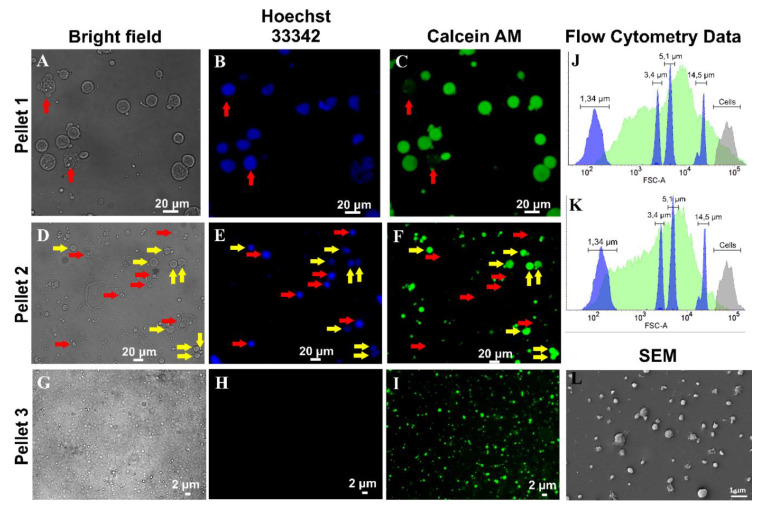
Characterization of CIMVs isolation by sequential centrifugation. Pellet 1—cell components obtained after the first step of centrifugation at 100g for 10 min; Pellet 2—cell components obtained after the second step of centrifugation at 300g for 20 min; Pellet 3—CIMVs obtained after the third step of centrifugation at 2000g for 25 min. **A**–**I**—fluorescence microscopy, images were captured using ZEISS Axio Observer Z1 microscope. J,K—Flow cytometry analysis. Blue—calibration particles (1.34, 3.4, 5.1, 14.5 μm), grey—native cells, green—pellet components. L—scanning electron microphotograph of CIMVs. ➨ karioplasts/nuclei; ➨ cells/cytoplasm-depleted cells.

**Figure 3 pharmaceutics-12-00577-f003:**
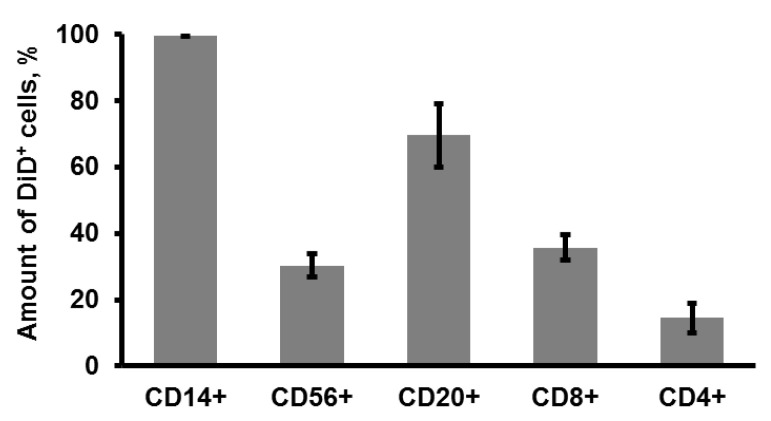
Cytochalasin B-induced membrane vesicles–mesenchymal stem cells (CIMVs-MSCs) uptake by leukocytes. CIMVs-MSCs (10 μg) stained with membrane dye DiD were incubated with peripheral blood mononuclear cells (PBMCs) for 24 h. PBMCs were incubated with antibodies to CD markers (anti-CD45, anti-CD3, anti-CD4, anti-CD8, anti-CD20, anti-CD14, and anti-CD56 monoclonal antibodies) and analyzed using flow cytometer BD FACS Aria III (BD Bioscience, USA). Experiments were repeated three times. The data represent mean ± SD.

**Figure 4 pharmaceutics-12-00577-f004:**
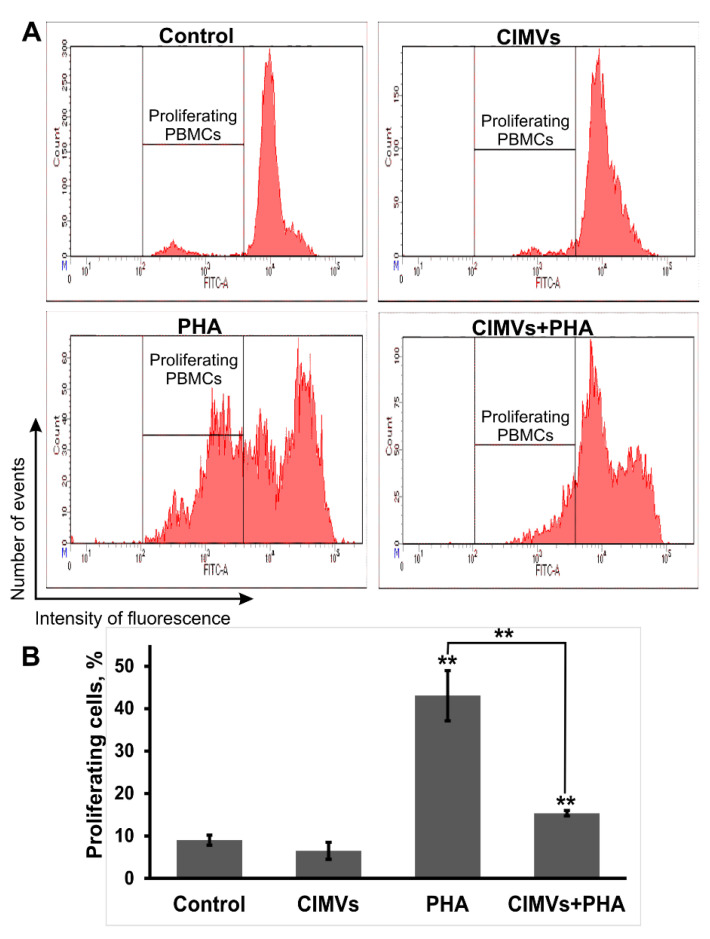
Analysis of phytohemagglutinin (PHA)-induced proliferation of PBMCs. PBMCs were stained with CFDA SE prior to incubation with PHA. Flow cytometry data (**A**). Histograms were generated using FACSDiva7 software (BD Bioscience, USA). The percent of cells with decreased CFDA SE fluorescence was taken to determine the PBMCs proliferation rate (**B**). The data represent mean ± SD. (*) Level of significance *p* < 0.05; (**) level of significance *p* < 0.01.

**Figure 5 pharmaceutics-12-00577-f005:**
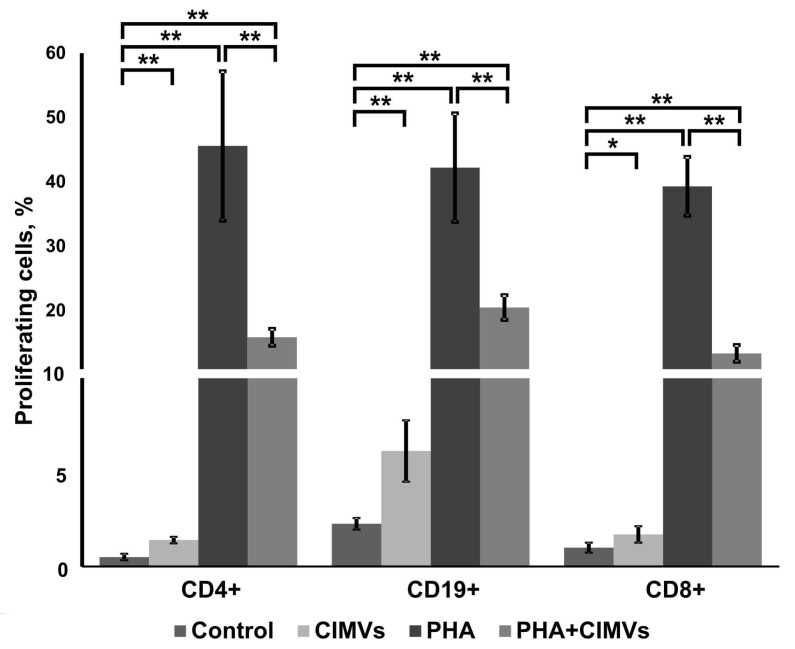
Analysis of the CIMVs-MSCs effect on PHA-induced proliferation of T-cytotoxic (CD8+), T-helper (CD4+), and B-cells (CD19+). Lymphocytes were stained with CFDA SE, followed by incubation with CIMVs-MSCs for 24 h and treatment with PHA (10 μg/mL). Immunostaining using anti-CD4, anti-CD19, and anti-CD8 monoclonal antibodies was done on the 3rd day after activation. The percent of proliferating cells was determined three days after PHA incubation using flow cytometer BD FACS Aria III (BD Bioscience, USA). Experiments were repeated three times. Data represent mean ± SD. (*) Level of significance *p* < 0.05; (**) level of significance *p* < 0.01.

**Figure 6 pharmaceutics-12-00577-f006:**
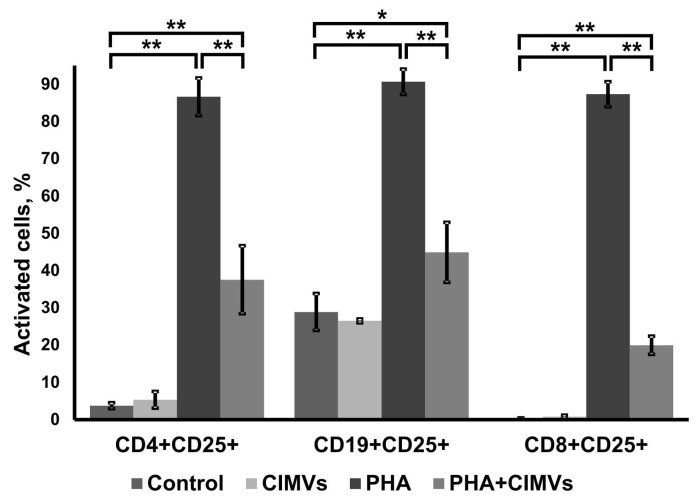
Effect of CIMVs-MSCs on lymphocyte activation. Lymphocytes were incubated with CIMVs-MSCs for 24 h followed by PHA activation (10 μg/mL). PBMCs were incubated with anti-CD4, anti-CD19, anti-CD8, and anti-CD25 monoclonal antibodies three days after PHA activation. The percent of activated cells was determined as CD25+ cells using flow cytometer BD FACS Aria III (BD Bioscience, USA). Experiments were repeated three times. The data represent mean ± SD. (*) Level of significance *p* < 0.05; (**) level of significance *p* < 0.01.

**Figure 7 pharmaceutics-12-00577-f007:**
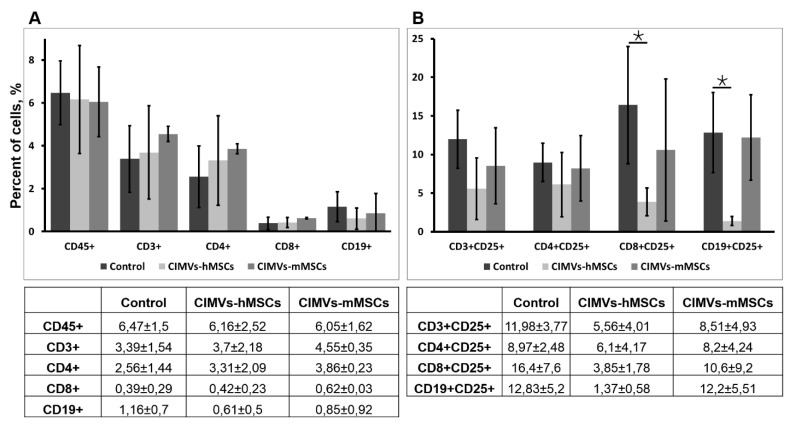
Immune response after CIMVs-hMSCs or CIMVs-mMSCs i.v. injection. Percent of the main populations of immune cells (**A**) and their activation status (**B**) were defined. Blood samples were collected 24 h after the i.v. injection with 15 µg of CIMVs-hMSCs or 15 µg of CIMVs-mMSCs. Percent of immune cells was determined using immunostaining and flow cytometry. The data represent mean ± SD.

**Table 1 pharmaceutics-12-00577-t001:** Cytokine analysis in culture medium of PBMCs treated with CIMVs-MSCs.

Cytokine	PBMCs	PBMCs+CIMVs-MSCs	CD4+ Cells	CD4+cells+ CIMVs-MSCs	CD8+ Cells	CD8+cells+ CIMVs-MSCs
EGF	89.01 ± 17.96	**60.31 ± 4.15**	9.91 ± 2.12	9.51 ± 1.15	10.56 ± 2.73	11.13 ± 0.66
FGF-2	9.81 ± 2.02	**2204.22 ±** **108.62**	<5.94	**2656.99 ±** **141.89**	<5.94	**2576.9 ±** **229.82**
Eotaxin (CCL11)	10.85 ± 1.9	11.34 ± 1.39	1.38	3.03 ± 0.37	<0.72	**2.1 ± 1.02**
TGF-a	7.07 ± 0.83	8.36 ± 0.51	<0.81	<0.81	<0.81	<0.81
G-CSF	2434.68 ± 313.12	**4823.67 ±** **1358.55**	3.29 ± 2.79	**28.53 ± 5.31**	<1.82	**73.76 ±** **15.37**
GM-CSF	66.63 ± 10.38	**109.33 ± 14.5**	1.13 ± 0.93	1.07 ± 0.05	2.51	3.57 ± 0.14
Fractalkine (CX3CL1)	43.73 ± 15.99	**16.79 ± 3.85**	<3.2	<3.2	<3.2	<3.2
IFN-a2	53.9 ± 24.33	84.06 ± 47.18	<1.27	<1.27	<1.27	<1.27
IL-10	3086.13 ± 112.82	3429.61 ± 576.28	<0.4	<0.4	<0.4	<0.4
MCP-3 (CCL7)	179.34 ± 22.65	**272.61 ±** **30.47**	<3.41	<3.41	<3.41	<3.41
IL-12p40	35.76 ± 6.05	41.58 ± 7.09	<1.39	<1.39	<1.39	<1.39
MDC (CCL22)	391.78 ± 70.17	**558.93 ± 70.78**	<13.47	<13.47	<13.47	<13.47
IL-12p70	2.33 ± 0.74	**5.73 ± 2.34**	<0.17	**0.24 ± 0.16**	<0.17	**0.31 ± 0.15**
PDGF-AA	144.38 ± 25.07	133.34 ± 20.2	10.68 ± 0.54	11 ± 1.15	8.53 ± 0.8	10.36 ± 1.56
IL-15	<1.03	1.44 ± 0.39	<1.03	<1.03	<1.03	<1.03
sCD40L	8.5 ± 3.82	6.21 ± 2.03	0.25 ± 0.19	0.57 ± 0.3	<LLOQ *	<LLOQ *
IL-1Ra	212.07 ± 56.48	228.47 ± 29.57	<0.85	<0.85	<0.85	<0.85
IL-1a	3541.85 ± 426.81	3755.48 ± 437.57	<3.2	<3.2	<3.2	<3.2
IL-9	1.07 ± 0.22	1.93 ± 1.37	<0.05	<0.05	<0.05	<0.05
IL-1b	2510.15 ± 254.06	**2913.38 ±** **120.82**	4.09 ± 0.71	4.9 ± 2.1	11 ± 1.36	**8.33 ± 1.15**
IL-4	124.01 ± 8.04	124.97 ± 5.77	<3.2	**9.05 ± 2.13**	<3.2	**6.7 ± 3.34**
IL-5	0.18 ± 0.06	0.45 ± 0.58	<LLOQ*	0.06 ± 0.01	0.02 ± 0.01	0.04 ± 0.02
IL-6	>7819.31	>7819.31	6.68 ± 3.3	**237.66 ±** **36.78**	9.32 ± 3.63	**359.11 ±** **0.39**
IL-7	55.8 ± 5.61	45.04 ± 17.6	<2.1	**4.91 ± 1.1**	<2.1	**8.19 ± 0.43**
IL-8	21222.36 ± 9057.91	>30603.7	25.82 ± 7.25	**440.74 ±** **86.82**	47.28 ± 26.46	**601.86 ±** **23.44**
IP10 (CXCL10)	51.66 ± 4.82	**65.36 ± 10.39**	4.08 ± 2.52	**9.9 ± 1.2**	2.67 ± 1.52	**35.78 ± 7.56**
MCP-1 (CCL2)	5640.54 ± 109.79	**6263.61 ±** **418.96**	6.83 ± 2.55	**52.27 ± 4.74**	<LLOQ*	**148.64 ±** **13.17**
MIP-1a (CCL3)	>9996.8	>9996.8	60.3 ± 12.85	60.72 ± 8.41	95.97 ± 20.97	81.65 ± 25.96
MIP-1b (CCL4)	7617.27 ± 1114.36	8438.14 ± 935.11	12.54 ± 3.65	12.07 ± 2.93	69.52 ± 0.22	61.76 ± 29.48
RANTES (CCL5)	>12068.63	>12068.63	152.43 ± 17.56	166.12 ± 18.63	131.65 ± 54.86	130.21 ± 27.08
TNFa	1825.11 ± 231.14	2237.44 ± 324.68	10.3 ± 3.97	11.12 ± 0.66	63.69 ± 10.27	55.52 ± 17.18

* Lower limit of quantification (LLOQ); Bold format indicates: *p*-value ≤ 0.05.

## Data Availability

All data generated or analyzed during this study are included in this published article. The data that support the findings of this study are available from the corresponding author upon request.

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
