# Peer review of "Mesenchymal Stem Cell Derived Biocompatible Membrane Vesicles Demonstrate Immunomodulatory Activity Inhibiting Activation and Proliferation of Human Mononuclear Cells"

_pharmaceutics, 2020, doi:10.3390/pharmaceutics12060577_

Round 1
Reviewer 1 Report
Authors described cytochalasin B to produce human mesenchymal stem cells derived cytochalasin B-induced membrane vesicles (CIMVs-MSCs), analyzed their effect on lymphocyte proliferation, activation, cytokine production in vitro and evaluate their immunogenicity in vivo. The research field is focused on CIMVs-MSCs, as authors have released lots of scientific reports to achieve the progress of technical procedure.
General experimental data would be reflecting some phenomena caused by CIMVs-MSCs; however, conclusion is not enough to argue that CIMVs-MSCs as a promising therapeutic instrument for the treatment of immune-mediated disorders due to various mechanisms underlying numerous immune disorders.
Authors carefully determined the processing of CIMVs-MSC in Figure 1 and 2; therefore, authors need to explain the superiority of CIMVs-MSCs as well as differences from extracellular vesicle (EV) and Exosome in introduction and discussion for readers.
The quality of MSCs used in the experiments should be explained before processing CIMVs-MSC. It exits concerns that ethical issue using human adipose tissue derived MSCs had been either purchased from a company or donor derived for experiments.
Critical points are that authors used peripheral blood mononuclear cells (PBMCs) with phytohemagglutinin (PHA). As PHA mainly stimulates CD4+T-cells, and less influence on CD8+ T-cells, PBMCs had been exposed with CIMVs-MSCs followed by PHA stimulation, then indirect effects on lymphocytes via monocytes and B-cells with uptake activity would be occurred, other than direct interaction of lymphocytes on CIMVs-MSCs.
As CD56 is mainly expressed on NK cells, but a minor population of CD8+ T cells also positive, CD3 negative CD56+ would be better to define NK cells. CD25+ lymphocytes are classified as activating T cells, B cells, and NK cells; on the other hand, CD25+CD4+ T cell population is also including regulatory T cells, which are important lymphocytes causing autoimmune disease.
Therefore, immune cells should be separated before each experiment for evaluating exact data. Proliferative activity and functional assays should be performed under sorted cell population like cytokine analysis provided in Table 1.
Authors need to evaluate the meaning of the cytokine profiles in Table 1 in Discussion, which are required for statistical evaluation.
Transplantation of allogeneic and xenogeneic CIMVs seems to be rough experiments as collateral evidence to explain in vivo analyses of CIMVs-MSCs.
Typographical quality including references is also required for better understanding of the report for readers as well as referees.
Author Response
We thank the reviewer for the precious and constructive comments and suggestions.
Comment:
Authors described cytochalasin B to produce human mesenchymal stem cells derived cytochalasin B-induced membrane vesicles (CIMVs-MSCs), analyzed their effect on lymphocyte proliferation, activation, cytokine production in vitro and evaluate their immunogenicity in vivo. The research field is focused on CIMVs-MSCs, as authors have released lots of scientific reports to achieve the progress of technical procedure.
General experimental data would be reflecting some phenomena caused by CIMVs-MSCs; however, conclusion is not enough to argue that CIMVs-MSCs as a promising therapeutic instrument for the treatment of immune-mediated disorders due to various mechanisms underlying numerous immune disorders.
Reply: As suggested, the Conclusion Section has been modified and rephrased as follow: “Obtained data can be used as a base for the developing the immunomodulation therapy using CIMVs-MSCs.”.
Comment:
Authors carefully determined the processing of CIMVs-MSC in Figure 1 and 2; therefore, authors need to explain the superiority of CIMVs-MSCs as well as differences from extracellular vesicle (EV) and Exosome in introduction and discussion for readers.
Reply: As suggested, we have discussed the differences of CIMVs from EV in the Introduction Section: “The main differences of CIMVs from EVs are: 1) more homogeneous content – the isolation method of CIMVs excludes the sorting of molecules; 2) CIMVs are produced from washed cells whereas EVs are isolated from body fluids or conditioned medium. CIMVs might imitate natural EVs in size and cytoplasmic membrane to serve as a vector for drugs and bioactive molecules delivery.”
Comment:
The quality of MSCs used in the experiments should be explained before processing CIMVs-MSC. It exits concerns that ethical issue using human adipose tissue derived MSCs had been either purchased from a company or donor derived for experiments.
Reply: As suggested, we have added the missing information in 2.4 Section: “MSCs were isolated from human adipose tissue. Signed informed consent was obtained from all donors. All experiments were carried out in accordance with an experimental protocol approved by the Biomedicine Ethic Expert Committee of Kazan Federal University and Republican clinical hospital (No. 218, 11.15.2012) based on article 20 of the Federal Legislation on "Health Protection of Citizens of the Russian Federation" № 323-FL, 21.11.2011.”
The MSCs quality is evaluated every time after the isolation procedure by immunostaining and flow cytometry. Human MSCs (passage 4) brightly expressing surface markers (CD90+, CD44+, CD29+, CD73+, STRO-1+) were chosen for the CIMVs production.
Comment:
Critical points are that authors used peripheral blood mononuclear cells (PBMCs) with phytohemagglutinin (PHA). As PHA mainly stimulates CD4+T-cells, and less influence on CD8+ T-cells, PBMCs had been exposed with CIMVs-MSCs followed by PHA stimulation, then indirect effects on lymphocytes via monocytes and B-cells with uptake activity would be occurred, other than direct interaction of lymphocytes on CIMVs-MSCs.
Reply: We agree that in PBMC there are an interplay between immune cell subsets, co-stimulation and signal transduction. We believe that experiments using PBMC culture are essential on the first stage of the investigation since PBMC culture retain immune cells interaction and imitate immune reaction in vitro. Detailed analysis of the effect of CIMVs-MSCs on isolated lymphocytes requires further investigation and is a focus of our future work.
Comment:
As CD56 is mainly expressed on NK cells, but a minor population of CD8+ T cells also positive, CD3 negative CD56+ would be better to define NK cells.
Reply: Thank you for this helpful suggestion. We have counted NK cells as CD3-CD56+ and corrected 3.2 Section.
Comment:
CD25+ lymphocytes are classified as activating T cells, B cells, and NK cells; on the other hand, CD25+CD4+ T cell population is also including regulatory T cells, which are important lymphocytes causing autoimmune disease.
Therefore, immune cells should be separated before each experiment for evaluating exact data. Proliferative activity and functional assays should be performed under sorted cell population like cytokine analysis provided in Table 1.
Reply: We agree that quantification of activated immune cells (CD25+ cells) included regulatory T cells which constitutively express CD25. We have now pointed on this circumstance in the Discussion section (Lines 396-398).
We would like to clarify this point since on the first stage of the investigation we have chosen the PBMC model system which reflects the complexity of the immune response in vivo. It is known that PHA activation protocol was developed for PBMC culture [1-4] and requires the presence of mixed population of immune cells [1].
However we fully agree that the effect of CIMVs on separated immune cells is of particular interest. We are working on the design of experiment and suggested considerations will be accounted for in future work.
Comment:
Authors need to evaluate the meaning of the cytokine profiles in Table 1 in Discussion, which are required for statistical evaluation.
Reply: At the request of the reviewer we have indicated the statistically significant changes (p-value ≤0.05) in cytokine profiles in Table 1. And we have added the meaning of p-value in the Discussion Section.
Comment:
Transplantation of allogeneic and xenogeneic CIMVs seems to be rough experiments as collateral evidence to explain in vivo analyses of CIMVs-MSCs.
Reply: It is known that low immunogenicity of MSCs [5] makes them an outstanding therapeutic instrument. However current understanding of the immune properties of CIMVs derived from MSCs remains limited. Therefore we conducted first steps in the investigation of immunogenicity of allogeneic and xenogeneic CIMVs in vivo.
Comment:
Typographical quality including references is also required for better understanding of the report for readers as well as referees.
Reply: As suggested by the reviewer through the manuscript the typos and grammar errors have been edited.
References:
- Ceuppens, J.L.; Baroja, M.L.; Lorre, K.; Van Damme, J.; Billiau, A. Human T cell activation with phytohemagglutinin. The function of IL-6 as an accessory signal. Journal of immunology 1988, 141, 3868-3874.
- Antas, P.R.; Oliveira, E.B.; Milagres, A.S.; Franken, K.C.; Ottenhoff, T.H.; Klatser, P.; Sarno, E.N.; Sampaio, E.P. Kinetics of T cell-activation molecules in response to Mycobacterium tuberculosis antigens. Memorias do Instituto Oswaldo Cruz 2002, 97, 1097-1099, doi:10.1590/s0074-02762002000800005.
- Zhao, W.; Gu, Y.H.; Song, R.; Qu, B.Q.; Xu, Q. Sorafenib inhibits activation of human peripheral blood T cells by targeting LCK phosphorylation. Leukemia 2008, 22, 1226-1233, doi:10.1038/leu.2008.58.
- Maccio, A.; Madeddu, C.; Chessa, P.; Panzone, F.; Lissoni, P.; Mantovani, G. Oxytocin both increases proliferative response of peripheral blood lymphomonocytes to phytohemagglutinin and reverses immunosuppressive estrogen activity. In vivo 2010, 24, 157-163.
- Saeedi, P.; Halabian, R.; Imani Fooladi, A.A. A revealing review of mesenchymal stem cells therapy, clinical perspectives and Modification strategies. Stem cell investigation 2019, 6, 34, doi:10.21037/sci.2019.08.11.

Reviewer 2 Report
This manuscript by Gomzikova et al isolates cytochalasin-B induced membrane vesicles (CIMVs) from primary human adipose-derived mesenchymal stem cells (MSCs) and uses them to assess their impact on peripheral blood mononuclear cells (PBMCs). CIMVs are similar to extracellular vesicles in that they are fractions of cellular membrane and cytoplasm, without nuclei, with the primary difference being they are artificially induced by destabilizing the actin cytoskeleton with cytochalasin-B. CIMVs-MSCs are hypothesized to have anti-inflammatory properties that may be useful for future use to treat auto-immune disorders and other chronic inflammatory diseases. In this manuscript, the authors use both primary MSCs and PBMCs to analyze the effect of CIMVs on proliferation and activation of various white blood cells populations. This manuscript does an excellent job describing their model system and methods. The data follow a logical progression of experiments. The English grammar could be improved in several areas prior to publication. There are two areas of concern for this manuscript:
- In Figures 5 and 6, the authors show data suggesting that CIMVs-MSCs can inhibit PHA-induced proliferation (Fig 5) and activation (Fig 6) of T (CD4+ and CD8+) and B cells. However, in figure 3 the authors had shown there is a wide range in the ability of cells to uptake CIMVs-MSCs based on acquired immunofluorescence from labeled CIMVs. It is unclear if the impact on proliferation and activation of lymphocytes is impacted by the frequency with which they are able to uptake CIMVs, or if this immunosuppressive effect occurs regardless of uptake efficiency. It would be beneficial if the authors could label the CIMVs with DiD then perform proliferation and CD25+ activation experiments on just the cells that uptake the CIMVs-MSCs.
- It would be helpful to the reader to identify the cytokines in Table 1 that had statistically significant differences between control and CIMVs-MSCs treated cells. A label within the Table (like * or underlining, etc), or extra columns with p values for each group would help.
Author Response
We thank the reviewer for the precious and constructive comments and suggestions.
Comment:
"This manuscript by Gomzikova et al isolates cytochalasin-B induced membrane vesicles (CIMVs) from primary human adipose-derived mesenchymal stem cells (MSCs) and uses them to assess their impact on peripheral blood mononuclear cells (PBMCs). CIMVs are similar to extracellular vesicles in that they are fractions of cellular membrane and cytoplasm, without nuclei, with the primary difference being they are artificially induced by destabilizing the actin cytoskeleton with cytochalasin-B. CIMVs-MSCs are hypothesized to have anti-inflammatory properties that may be useful for future use to treat auto-immune disorders and other chronic inflammatory diseases.
In this manuscript, the authors use both primary MSCs and PBMCs to analyze the effect of CIMVs on proliferation and activation of various white blood cells populations. This manuscript does an excellent job describing their model system and methods. The data follow a logical progression of experiments. The English grammar could be improved in several areas prior to publication. There are two areas of concern for this manuscript:
In Figures 5 and 6, the authors show data suggesting that CIMVs-MSCs can inhibit PHA-induced proliferation (Fig 5) and activation (Fig 6) of T (CD4+ and CD8+) and B cells. However, in figure 3 the authors had shown there is a wide range in the ability of cells to uptake CIMVs-MSCs based on acquired immunofluorescence from labeled CIMVs. It is unclear if the impact on proliferation and activation of lymphocytes is impacted by the frequency with which they are able to uptake CIMVs, or if this immunosuppressive effect occurs regardless of uptake efficiency. It would be beneficial if the authors could label the CIMVs with DiD then perform proliferation and CD25+ activation experiments on just the cells that uptake the CIMVs-MSCs."
Reply: Thanks for the constructive advice concerning the analysis of frequency of CIMVs uptake and its impact on proliferation and activation of lymphocytes. We have conducted the required experiment. CIMVs were prestained with DiD dye (as described in 2.8 Section) and added to the PBMC culture (in concentration 10μg CIMVs per 1.5х105 cells). After 24 hours PBMCs were activated by incubation with 10 μg/ml PHA for 3 days. Then the proliferation of lymphocytes were determined.
We found that 48,7±3,5% of native CD8-positive cells captured CIMVs-MSCs, whereas PHA activation led to 99,76±0,11% of CD8-positive cells captured CIMVs-MSCs. Among native CD4-positive cells 40±11,85% of cells captured CIMVs-MSCs, and 99,8±0,058 of PHA-activated CD4-positive cells captured CIMVs-MSCs. And 91,5±4,8% of native B-cells captured CIMVs-MSCs, 99,6±0,4% of PHA-activated CD20-positive cells contained CIMVs-MSCs (Sup.Fig.1). We observed that PHA activation led to increasing of CIMVs capturing by lymphocytes (Sup.Fig.1).
Supplemental Figure 1. CIMVs-MSCs uptake by lymphocytes. CIMVs-MSCs (10 μg) stained with membrane dye DiD were incubated with PBMCs for 24 hours, followed by treatment with PHA (10 μg/ml). PBMCs were analyzed using flow cytometer BD FACS Aria III (BD Bioscience, USA). The data represents mean ± SD.
Then we have investigated the PHA-activated proliferation of lymphocytes, containing CIMVs-MSCs. We found that CIMVs-MSCs inhibit PHA-activated proliferation of lymphocytes (Sup.Fig.2). CIMVs inhibited PHA-activated proliferation of CD8+, CD4+ and CD20+ in 1.7, 2,9 and 3,7 times respectively.
Supplemental Figure 2. CIMVs-MSCs effect on PHA induced proliferation of T-cytotoxic (CD8+), T-helper (CD4+) and B-cells (CD20+). Lymphocytes were stained with CFDA SE, followed by incubation with DiD-labelled CIMVs-MSCs for 24 hours and treatment with PHA (10 μg/ml). The percent of proliferating cells was evaluated 3 days after PHA incubation. Data represents mean ± SD.
We have confirmed our previous results and showed that PHA activation led to increasing CIMVs uptake efficiency by lymphocytes. Due to almost all PHA-activated lymphocytes contained CIMVs (in average 99,7% of uptake efficiency), we believe that is reasonable to conclude that CIMVs impact on PHA-activated proliferation and activation of lymphocytes. As suggested we have discussed this findings in the Results and Discussion Sections.
Comment:
It would be helpful to the reader to identify the cytokines in Table 1 that had statistically significant differences between control and CIMVs-MSCs treated cells. A label within the Table (like * or underlining, etc), or extra columns with p values for each group would help.
Reply: As suggested we have indicated the statistically significant changes (p-value ≤0.05) in cytokine profiles in Table 1. And we have added the meaning of p-value in the Discussion Section.
Please see the attached file, which contains the Sup.Fig. 1-2.

Reviewer 3 Report
The manuscript presented by M.O.Gomzikova was aimed to investigate the immunological properties of CIMVs-MSC and their effect on human PBMCs. The study design was appropriate and the manuscript was well written with the following comments:
Line 87 …using FACS. PBMCs (1X106 cells/ml) were stained… forgot the superscript in cell number?
Line 328 I found Figure 7A in this section is confusing. According to the text, murine PBMC was purified by gradient density separation and stained with mAbs for FACS. However the percent of cells, especially the % of CD45+ cells in murine blood seemed very low to me. A gating strategy for the FACS analysis will be helpful for the reader to understand. Or at least specify the parent gates. Also in the same section, authors only look into the overnight effect of injected CIMVs-hMSCs and CIMVs-mMSCs on murine immune cells proliferation. However, to my understanding, significant cell proliferation observed overnight is not usual nor the upregulation of CD25+ in T cells. Thus not supportive to the statement that the CIMVs are non-immunogenic in mice.
Author Response
Comment:
The manuscript presented by M.O.Gomzikova was aimed to investigate the immunological properties of CIMVs-MSC and their effect on human PBMCs. The study design was appropriate and the manuscript was well written with the following comments:
Line 87 …using FACS. PBMCs (1X106 cells/ml) were stained… forgot the superscript in cell number?
Reply: We thank the reviewer for the positive comment. As suggested we have edited the typos.
Comment:
Line 328 I found Figure 7A in this section is confusing. According to the text, murine PBMC was purified by gradient density separation and stained with mAbs for FACS. However the percent of cells, especially the % of CD45+ cells in murine blood seemed very low to me. A gating strategy for the FACS analysis will be helpful for the reader to understand. Or at least specify the parent gates. Also in the same section, authors only look into the overnight effect of injected CIMVs-hMSCs and CIMVs-mMSCs on murine immune cells proliferation. However, to my understanding, significant cell proliferation observed overnight is not usual nor the upregulation of CD25+ in T cells. Thus not supportive to the statement that the CIMVs are non-immunogenic in mice.
Reply: As suggested we have added the information about the gating strategy for the FACS analysis in Supplementary Data (Sup.Fig.3).
We chose 24 hours to evaluate the immune response based on the findings of Copeland Sh. et al. The authors showed that proinflammatory cytokines peaked around 2 h after endotoxin infusion in mice, elevation in core temperature peaks around 3 h after, hematological response appeared 4 h after endotoxin infusion in mice and 6 to 9 h after endotoxin infusion in humans [6]. Yoon H. et al. showed that CD8+ T cells could divide and proliferate with an initial cell division time of as short as 2 hours [7]. Collectively, these studies indicates that immune response are developed during first few hours.
We are fully agree that immune cell proliferation might be detected 3–7 days following primary immunization. And detailed analysis of immune response on transplantation of allogeneic and xenogeneic CIMVs in mice requires further investigation and is a focus of our future work.
To point out this limitation of the study we have edited the conclusion (by removing the statement that the CIMVs are non-immunogenic in mice) and added the following text in the Discussion Section: “However further research is needed with a longer observation time”.
References:
6. Copeland, S.; Warren, H.S.; Lowry, S.F.; Calvano, S.E.; Remick, D.; Inflammation; the Host Response to Injury, I. Acute inflammatory response to endotoxin in mice and humans. Clinical and diagnostic laboratory immunology 2005, 12, 60-67, doi:10.1128/CDLI.12.1.60-67.2005.
7. Yoon, H.; Kim, T.S.; Braciale, T.J. The cell cycle time of CD8+ T cells responding in vivo is controlled by the type of antigenic stimulus. PloS one 2010, 5, e15423, doi:10.1371/journal.pone.0015423.
Reviewer 4 Report
The manuscript describes an inhibitory effect of CIMVs-MSCs on human PBMC proliferation and activation. Moreover, upon treatment with CIMVs-MSCs, production of cytokines was assessed on total PBMCs and isolated CD4+ and CD8+ T cells . However, It has some concerns that should be better addressed:
- Authors claim that monocytes and B cells preferably uptake CIMVs-MSCs compared to T cells. Since ,in Fig.3, it has been shown co-espression of DiD+ cells with the indicated populations, can authors discriminate if CIMVs-MSCs are outside or inside the populations analysed? Moreover, it is well known the role of dendritic cells in orchestrating immune response. Considering the aim of this study, why authors excluded this population from the analysis?
- Inhibition of PBMC proliferation is a key result of this study, but in Fig.5 authors show that, in the absence of PHA stimulation, treatment with CIMVs-MSCs can increase the percentage of proliferating cells? do authors have some explanation for this discrepancy?
- Concerning cytokine assay: the choice to show data using table make results more difficult to read. Moreover, increase as well as decrease observed in the relative cytokines are statistically different? Considering that the totality of monocytes can uptake CIMVs-MSCs and their role in driving T cell response, why authors did'nt isolate also monocytes for multiplex analysis? Beside cytokine analysis, to make more clear the immunomodulatory activity of CIMVs-MSCs, it could be useful evaluate polarization of CD4+ T cell, by analysing trascription factor expression, upon treatment with CIMV-MSCs or eventually upon co-culture with CIMVs-MSCs stimulated monocytes.
- Considering data obtained from multiplex analysis, do authors have some hypothesis about the mechanism underlying inhibition of proliferation induced by CIMVs-MSCs?
Author Response
We thank the reviewer for the precious and constructive comments and suggestions.
Comment:
The manuscript describes an inhibitory effect of CIMVs-MSCs on human PBMC proliferation and activation. Moreover, upon treatment with CIMVs-MSCs, production of cytokines was assessed on total PBMCs and isolated CD4+ and CD8+ T cells . However, It has some concerns that should be better addressed:
Authors claim that monocytes and B cells preferably uptake CIMVs-MSCs compared to T cells. Since ,in Fig.3, it has been shown co-espression of DiD+ cells with the indicated populations, can authors discriminate if CIMVs-MSCs are outside or inside the populations analysed? Moreover, it is well known the role of dendritic cells in orchestrating immune response. Considering the aim of this study, why authors excluded this population from the analysis?
Reply: As suggested we have analyzed co-expression of DiD stained CIMVs with immune cells populations using confocal microscopy. We found that DiD stained CIMVs are mostly inside of analyzed cells (Sup.Fig.2,3). This is most clearly seen in CD14-positive cells, due to their relatively large diameter (Sup.Fig.3). Co-localization of DID and AB-staining signals are pointed with arrows. We have added this information in the Results section (Line 236).
In our work we investigated the major populations of immune cells as a first step in the evaluation of immunomodulatory activity of CIMVs. We are working on the design of experiment to identify the molecular mechanisms underlaying of CIMVs immunosuppression and suggested considerations will be accounted.
Comment:
Inhibition of PBMC proliferation is a key result of this study, but in Fig.5 authors show that, in the absence of PHA stimulation, treatment with CIMVs-MSCs can increase the percentage of proliferating cells? do authors have some explanation for this discrepancy?
Reply: Thank you for highlighting this phenomenon. As suggested, we have added the missing discussion. Previously Rasmusson I. et al. demonstrated that MSCs have a stimulatory effect on human B cells, showing that MSC have the ability to stimulate or suppress Ab secretion depending on the level of stimulus used to trigger B cells [8]. It was shown that MSC possess two distinctive activities - MSC are able to support and suppress allogeneic lymphocytes responses, depending on their concentrations (MSC:CD3+ ratios). The authors showed that soluble factors particularly IL-6 secreted by MSC mediate this effect [9]. We have discussed this phenomenon in the Discussion section (Lines 422-425).
Comment:
Concerning cytokine assay: the choice to show data using table make results more difficult to read. Moreover, increase as well as decrease observed in the relative cytokines are statistically different?
Considering that the totality of monocytes can uptake CIMVs-MSCs and their role in driving T cell response, why authors did'nt isolate also monocytes for multiplex analysis? Beside cytokine analysis, to make more clear the immunomodulatory activity of CIMVs-MSCs, it could be useful evaluate polarization of CD4+ T cell, by analysing trascription factor expression, upon treatment with CIMV-MSCs or eventually upon co-culture with CIMVs-MSCs stimulated monocytes.
Reply: At the request of the reviewer we have indicated the statistically significant changes (p-value ≤0.05) in cytokine profiles in Table 1. And we have added the meaning of p-value in the Discussion Section.
We focused on CD4, CD19, CD8-positive cells populations to comprehensively analyze not only cytokine secretion profile but their proliferation and activation. The suggested considerations regarding the cytokine secretion profile of monocytes will be accounted in our future work. Thank you for this helpful suggestion.
Thank you. We can clarify this point. We analyzed CD4+ T cells, including Th1,Th2, Th17 and T-reg subsets by the cytokine profiles (Table 1). It is known that CD4+ T cell categorized by the cytokines that they produce (so called “signature cytokines”) [10]. We found that CIMVs did not influence on the expression of IL-15, GM-CSF and TNF-a – cytokines produced by Th1 cells. Therefore CIMVs-MSCs did not induce Th1 activation and inflammation. We have added this information in the Discussion section (Lines 419-424). We agree that investigation of transcription factor expression, upon co-culture with CIMV-MSCs stimulated monocytes is of particular interest. We are planning to conduct this experiments in the future.
Comment:
Considering data obtained from multiplex analysis, do authors have some hypothesis about the mechanism underlying inhibition of proliferation induced by CIMVs-MSCs?
Reply: As we have mentioned above treatment with CIMVs did not influence on the expression of IL-15, GM-CSF and TNF-a in CD4+ cells– cytokines produced by Th1 cells. Whereas CIMVs induced secretion of G-CSF (p=0.021) (Table 1). Our data corroborate results published by Rozenberg A. et al. who showed that hMSCs Inhibit Th1 Responses yet Induce Th17 Responses [11]. We believe that CIMVs-MSCs might partly imitate the biological activity of MSCs for which exert both anti-inflammatory and proinflammatory effects is characteristic. As suggested we have added this information in the Discussion Section (Lines 419-424).
References:
8. Rasmusson, I.; Le Blanc, K.; Sundberg, B.; Ringden, O. Mesenchymal stem cells stimulate antibody secretion in human B cells. Scandinavian journal of immunology 2007, 65, 336-343, doi:10.1111/j.1365-3083.2007.01905.x.
9. Najar, M.; Rouas, R.; Raicevic, G.; Boufker, H.I.; Lewalle, P.; Meuleman, N.; Bron, D.; Toungouz, M.; Martiat, P.; Lagneaux, L. Mesenchymal stromal cells promote or suppress the proliferation of T lymphocytes from cord blood and peripheral blood: the importance of low cell ratio and role of interleukin-6. Cytotherapy 2009, 11, 570-583, doi:10.1080/14653240903079377.
10. Dittel, B.N. CD4 T cells: Balancing the coming and going of autoimmune-mediated inflammation in the CNS. Brain, behavior, and immunity 2008, 22, 421-430, doi:10.1016/j.bbi.2007.11.010.
11. Rozenberg, A.; Rezk, A.; Boivin, M.N.; Darlington, P.J.; Nyirenda, M.; Li, R.; Jalili, F.; Winer, R.; Artsy, E.A.; Uccelli, A., et al. Human Mesenchymal Stem Cells Impact Th17 and Th1 Responses Through a Prostaglandin E2 and Myeloid-Dependent Mechanism. Stem cells translational medicine 2016, 5, 1506-1514, doi:10.5966/sctm.2015-0243.
Round 2
Reviewer 1 Report
Authors clearly revised the manuscript as reviewers' concern and suggestion for exact translation of the immunological analyses.
We hope for authors to perform single cell-RNA sequence for the understanding of molecular mechanisms underlying the phenomena in future.